TECHNICAL RELEASE

# NucBalancer: streamlining barcode sequence selection for optimal sample pooling for sequencing

Saurabh Gupta[1,2,*] and Ankur Sharma[1,2,3,*]

1 Harry Perkins Institute of Medical Research, QEII Medical Centre and Centre for Medical Research, 6 Verdun Street, Nedlands, Perth, Western Australia, 6009, Australia

2 Curtin Medical School, Curtin Health Innovation Research Institute (CHIRI), Curtin University, Perth, Western Australia, 6102, Australia

3 Translational Genomics Program, Garvan Institute of Medical Research and Kinghorn Cancer Centre, Darlinghurst, New South Wales, 2010, Australia

## ABSTRACT

Recent advancements in next-generation sequencing (NGS) technologies have brought to the forefront the necessity for versatile, cost-effective tools capable of adapting to a rapidly evolving landscape. The emergence of numerous new sequencing platforms, each with unique sample preparation and sequencing requirements, underscores the importance of efficient barcode balancing for successful pooling and accurate demultiplexing of samples. Recently launched new sequencing systems claiming better affordability comparable to more established platforms further exemplifies these challenges, especially when libraries originally prepared for one platform need conversion to another. In response to this dynamic environment, we introduce NucBalancer, a Shiny app developed for the optimal selection of barcode sequences. While initially tailored to meet the nucleotide, composition challenges specific to G400 and T7 series sequencers, NucBalancer's utility significantly broadens to accommodate the varied demands of these new sequencing technologies. Its application is particularly crucial in single-cell genomics, enabling the adaptation of libraries, such as those prepared for 10x technology, to various sequencers including G400 and T7 series sequencers. NucBalancer efficiently balances nucleotide composition and sample concentrations, reducing biases and enhancing the reliability of NGS data across platforms. Its adaptability makes it invaluable for addressing sequencing challenges, ensuring effective barcode balancing for sample pooling on any platform.

**Availability and implementation:** NucBalancer is implemented in R and is available at https://github.com/ersgupta/NucBalancer. Additionally, a shiny interface is available at https://ersgupta.shinyapps.io/NucBalancer/.

**Submitted:** 19 April 2024

* Corresponding authors. E-mail: saurabh.gupta@curtin.edu.au; ankur.sharma@garvan.org.au

Preprint submitted at https://doi.org/10.1101/2024.09.06.611747

**Subjects** Software and Workflows, Genetics and Genomics, Bioinformatics

## INTRODUCTION

The field of genomics has been profoundly transformed by the evolution of next-generation sequencing (NGS) technologies. While Illumina (NovaSeq, NextSeq, and MiSeq) and MGI (G400 and T7) sequencing platforms have emerged as prominent tools, the landscape is now further enriched by the introduction of additional sequencing platforms by Element Biosciences (AVITI System) and Ultima Genomics (UG100). These new entrants are rapidly

gaining attention, each offering unique advantages that contribute to the diversity and capability of genomic research tools available.

Illumina, renowned for its high-quality output, has long set a standard in the field. Illumina sequencing chemistry employs a sequencing-by-synthesis (SBS) approach, where fluorescently labeled nucleotides are incorporated into the DNA strand and imaged to determine the sequence. MGI utilizes combinatorial probe-anchor synthesis (cPAS) chemistry, which differs from Illumina's SBS by using pre-labeled probes and combinatorial anchoring. Few recent studies have highlighted MGI's sequencers as alternatives to Illumina platforms [1, 2]. The cost-effectiveness is not merely a financial advantage; it enables researchers to study more samples and larger cohorts, thereby enhancing the scope and scale of genomic research. The new sequencing platforms, represented by AVITI System and UG100, also work on SBS chemistries and further expand these possibilities. They are being recognized for their unique features, such as increased throughput, specialized applications, or even greater cost efficiencies, thus providing a wider range of options for genomic studies. This progression provides a valuable opportunity for more accessible genomic research, especially in resource-limited settings where the ability to analyze larger datasets at reduced costs can significantly advance scientific understanding and discovery.

A pivotal aspect of optimizing these technologies lies in the preparation and pooling of multiple samples, necessitating the use of precisely balanced barcode sequences. In particular, it is essential to ensure that the nucleotide composition at each position in the barcode sequences is balanced, minimizing biases that may arise due to sequencing errors or base-calling artifacts [3]. While this is a critical step in SBS approaches, it becomes even more crucial when converting libraries for use with cPAS-based sequencers. The challenge intensifies due to the distinct sequencing chemistries and pooling requirements of these platforms, making the selection of appropriate barcode sequences a significant hurdle. Although there have been some attempts to develop tools for this purpose [4–7], they often lack the flexibility needed to account for extensive customization, such as varying sample concentrations and experimental constraints.

In response to these challenges, we present NucBalancer, an R-based Shiny application. NucBalancer is designed to facilitate sample multiplexing for various sequencing technologies, including the conversion of libraries prepared for one sequencing chemistry to be compatible with another. It serves as a versatile tool for generating base-balanced barcodes suitable for any sequencing platform. It particularly addresses the needs of cost-efficient single-cell omics studies, offering a solution for efficiently balancing nucleotide compositions and meeting the unique requirements of different sequencing methodologies. By streamlining the barcode selection process, NucBalancer ensures balanced and accurate demultiplexing across various platforms, thereby enhancing the reliability and fidelity of NGS data in a broad spectrum of genomic research.

## RESULTS AND IMPLEMENTATION

NucBalancer is developed in R, featuring a user-friendly Shiny interface for ease of use. The source code is available on GitHub (https://github.com/ersgupta/NucBalancer). The tool requires only one additional package, "optparse", and relies solely on utilities from the base R package, making it compatible with all R versions. A brief workflow of the methodology is provided in Figure 1.

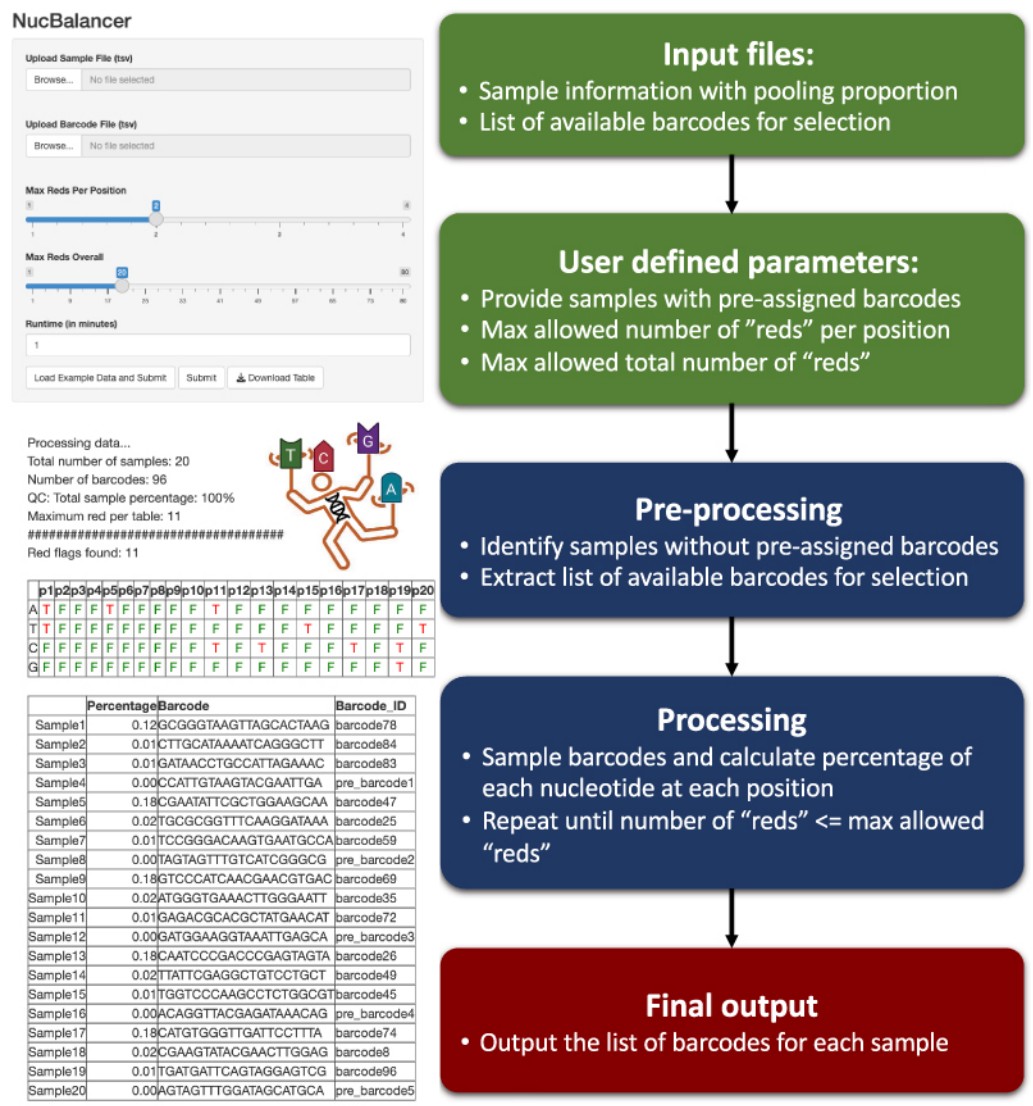

**Figure 1.  An overview of NucBalancer workflow.**
An illustration depicting the different steps in NucBalancer and snapshots of the input and output from the shiny app of NucBalancer.

The success of a nucleotide pooling strategy relies on achieving an equitable distribution of the four possible nucleotides (A, T, C, and G) at each position in the sequence. In an ideal scenario, an even 25% representation of each nucleotide would ensure reliable results. However, to accommodate natural variation and potential experimental deviations, a practical range of 12.5% to 67.5% for each nucleotide is considered acceptable. Also, this is the recommended range for sequencing on MGI platforms.

To assess the quality of a pooling configuration, we employ a systematic evaluation approach. We examine each nucleotide's percentage representation at every position within the sequence. If the observed percentage for any nucleotide falls outside the acceptable range, it raises a "red" flag, indicating a potential issue with the pooling arrangement at that

**Table 1.** Comparison of performance of the existing tools and NucBalancer to identify optimal barcodes for the example dataset available on GitHub.

| Tool | WebApp | Includes pooling percentage as input? | Number of bases outside the range of 12.5% to 67.5% |
|---|---|---|---|
| NucBalancer | Yes | Yes | 5 |
| DNABarcodes | No | No | 17 |
| BARCOSEL | Yes | No | 14 |
| DNABarcodeCompatibility | Yes | No | NA* |
| Checkmyindex | Yes | No | 23 |

\* DNABarcodeCompatibility does not support >6 plex and the test data contains 20 samples.

specific position. In essence, a red flag signifies a deviation from the desirable nucleotide distribution, which could lead to biased results in downstream analyses.

To provide flexibility in quality control, our tool enables users to define two critical parameters: the maximum number of red flags for the entire pooling set and the maximum number of red flags permissible at any single position. These parameters empower users to balance the trade-off between stringency and practicality, ensuring the optimal performance of their nucleotide pooling strategy. Table 1 shows a comparison of the performance of finding optimal barcodes for the test data set available on GitHub. The previously available tools do not consider the non-uniform pooling of libraries, which is often especially useful when pooling libraries which need low and high depth. Hence, these tools fail to find the optimal set of barcodes to pool such a sample set.

In summary, our tool employs a comprehensive assessment mechanism to gauge the adherence of a nucleotide pooling set to the desired nucleotide distribution range. By offering customizable criteria for red flag thresholds, users can tailor the pooling strategy to their specific experimental context, optimizing accuracy while accommodating natural variations.

## CONCLUSION

We present NucBalancer, a Shiny app designed for the optimal selection of barcode sequences for sample multiplexing in sequencing. Its user-friendly interface makes it accessible to both bioinformaticians and experimental researchers, enhancing its utility in adapting libraries prepared for one sequencing platform to be compatible with others. NucBalancer's incorporation of dynamic parameters, including customizable red flag thresholds, allows for precise and practical barcode sequencing strategies. This adaptability is key in ensuring uniform nucleotide distribution, particularly in MGI sequencing and single-cell genomic studies, leading to more reliable and cost-effective sequencing outcomes across various experimental conditions.

## DATA AVAILABILITY

Test data is available in GitHub. Snapshots of this data and code are available in Software Heritage [8] (Figure 2).

## AVAILABILITY AND REQUIREMENTS

- Project name: NucBalancer
- Project home page: https://github.com/ersgupta/NucBalancer
- Operating system(s): Platform independent
- Programming language: R

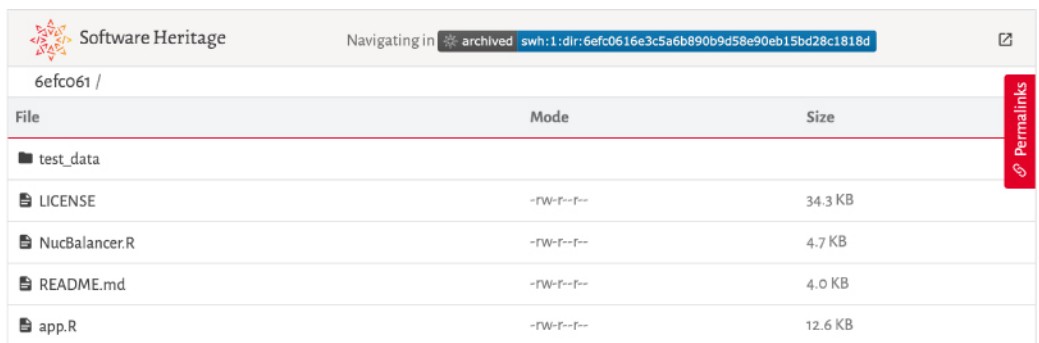

**Figure 2.** The Software Heritage archive of the NucBalancer code [8].
https://archive.softwareheritage.org/browse/embed/swh:1:dir:6efc0616e3c5a6b890b9d58e90eb15bd28c1818d;
origin=https://github.com/ersgupta/NucBalancer;visit=swh:1:snp:ba83cee787e227fffb8fbf2bf8321619b05bd60e;
anchor=swh:1:rev:7c444134eeedac12c828652a2f9218128b7c3d4d/

- License: GPL-3.0
- RRID:SCR_025845.

The source code for NucBalancer is available at https://github.com/ersgupta/NucBalancer. The tool is made available by a user-friendly shiny app at https://ersgupta.shinyapps.io/NucBalancer/.

## ABBREVIATIONS

cPAS: combinatorial probe-anchor synthesis; NGS: Next Generation Sequencing; SBS: sequencing-by-synthesis.

## DECLARATIONS

### Ethics approval and consent to participate

Not applicable.

### Competing interests

The authors declare that they have no competing interests.

### Consent for publication

Not applicable.

### Funding

This work was funded by the National Health and Medical Research Council (NHMRC) through the Ideas Grant [2021/GNT2010795] and Perkins-Curtin Start-up Fellowship awarded to AS.

### Acknowledgements

This work was supported by resources provided by the Pawsey Supercomputing Research Centre with funding from the Australian Government and the Government of Western Australia. The authors thank Jennifer Currenti and Rhea Pai for useful discussions.

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
