## [Editor Report]

Editor’s AssessmentThis paper presents NucBalancer, a R-pipeline and Shiny app designed for the optimal selection of barcode sequences for sample multiplexing in sequencing. Providing a user-friendly interface aiming to make this process accessible to both bioinformaticians and experimental researchers, enhancing its utility in adapting libraries prepared for one sequencing platform to be compatible with others. Important now with the introduction of additional sequencing platforms by Element Biosciences (AVITI System) and Ultima Genomics (UG100) increasing the diversity and capability of genomic research tools available. NucBalancer’s incorporation of dynamic parameters, including customizable red flag thresholds, allows for precise and practical barcode sequencing strategies. This adaptability is key in ensuring uniform nucleotide distribution, particularly in MGI sequencing and single-cell genomic studies, leading to more reliable and cost-effective sequencing outcomes across various experimental conditions. All the code is available under an open source license, and upon review the authors have also shared the code for the Shiny app.Editor’s AssessmentThis paper presents NucBalancer, a R-pipeline and Shiny app designed for the optimal selection of barcode sequences for sample multiplexing in sequencing. Providing a user-friendly interface aiming to make this process accessible to both bioinformaticians and experimental researchers, enhancing its utility in adapting libraries prepared for one sequencing platform to be compatible with others. Important now with the introduction of additional sequencing platforms by Element Biosciences (AVITI System) and Ultima Genomics (UG100) increasing the diversity and capability of genomic research tools available. NucBalancer’s incorporation of dynamic parameters, including customizable red flag thresholds, allows for precise and practical barcode sequencing strategies. This adaptability is key in ensuring uniform nucleotide distribution, particularly in MGI sequencing and single-cell genomic studies, leading to more reliable and cost-effective sequencing outcomes across various experimental conditions. All the code is available under an open source license, and upon review the authors have also shared the code for the Shiny app.

---

## [Reviewer Report]

Indicate in the comments box below whether you are happy with the changes made or if the manuscript is unacceptable.Comments on revised manuscriptI thank the authors for the improvements they made on this new version of the manuscript. At this stage, I'm not totally satisfied for the following reasons: - authors tell the source code of the Shiny app is now available on GitHub, but I have not been able to find it. - in the manuscript, the sentence "The tool does not have any dependency other than the utilities from the base R package" is no longer true as the tool now uses optparse. - in table 1, checkMyIndex is referenced with no web interface available white it actually exists (https://checkmyindex.pasteur.fr/). Moreover, the proposed web interface could still be improved. For instance: - it would be great to add something to show the algorithm is currently looking for a solution. - check the input files have a valid structure to be used. - display the input files when they are loaded to make sure the user uploaded the correct file.

---

## [Reviewer Report]

Indicate in the comments box below whether you are happy with the changes made or if the manuscript is unacceptable.Comments on revised manuscriptThe authors have addressed all my concerns.

---

## [Reviewer Report]

Reviewer name and names of any other individual's who aided in reviewerAamir W. KhanDo you understand and agree to our policy of having open and named reviews, and having your review included with the published manuscript. (If no, please inform the editor that you cannot review this manuscript.)YesIs the language of sufficient quality?YesPlease add additional comments on language quality to clarify if neededIs there a clear statement of need explaining what problems the software is designed to solve and who the target audience is? YesAdditional CommentsThe tool has novel features not reported in previous tools for barcoding.Is the source code available, and has an appropriate Open Source Initiative license <a href="https://opensource.org/licenses" target="_blank">(https://opensource.org/licenses)</a> been assigned to the code?YesAdditional CommentsThe tool is available as an R script as well as a shiny app.As Open Source Software are there guidelines on how to contribute, report issues or seek support on the code?YesAdditional CommentsIs the code executable?YesAdditional CommentsI tested the R code and it works well as per the instructions.Is installation/deployment sufficiently outlined in the paper and documentation, and does it proceed as outlined?YesAdditional CommentsI would suggest mentioning a few features that are novel or superior to other tools. Perhaps adding a table specifying these novel features that are not part of existing tools will add value to MS.Is the documentation provided clear and user friendly?YesAdditional CommentsThe documentation is provided in a clear and user-friendly way. The input file formats are given in the GitHub page.  It would be better to add an example to the shiny app page.Is there enough clear information in the documentation to install, run and test this tool, including information on where to seek help if required?YesAdditional CommentsThe documentation is given with clarity.Is there a clearly-stated list of dependencies, and is the core functionality of the software documented to a satisfactory level?YesAdditional CommentsDependencies are mentioned on the tool documentation page and can be installed if R is already installed.Have any claims of performance been sufficiently tested and compared to other commonly-used packages? Not applicableAdditional CommentsIs test data available, either included with the submission or openly available via cited third party sources (e.g. accession numbers, data DOIs)?YesAdditional CommentsAre there (ideally real world) examples demonstrating use of the software? YesAdditional CommentsIs automated testing used or are there manual steps described so that the functionality of the software can be verified?YesAdditional CommentsAny Additional Overall Comments to the AuthorThe authors have a well-written MS describing the NucBalancer tool. The tool adds value for sequencing by pooling samples and will be useful as we make technological advancements in the sequencing space.RecommendationAccept

---

## [Reviewer Report]

Reviewer name and names of any other individual's who aided in reviewerHugo VaretDo you understand and agree to our policy of having open and named reviews, and having your review included with the published manuscript. (If no, please inform the editor that you cannot review this manuscript.)YesIs the language of sufficient quality?YesPlease add additional comments on language quality to clarify if neededIs there a clear statement of need explaining what problems the software is designed to solve and who the target audience is? YesAdditional CommentsThe manuscript explains the constraints to be satisfied when looking for barcodes but more details about the context (Illumina chemistry for instance) would be appreciated. Moreover, is the software compatible with dual-indexing?Is the source code available, and has an appropriate Open Source Initiative license <a href="https://opensource.org/licenses" target="_blank">(https://opensource.org/licenses)</a> been assigned to the code?YesAdditional CommentsThe source code of the program is available on GitHub as a R script, but the source code of the Shiny application is not available.As Open Source Software are there guidelines on how to contribute, report issues or seek support on the code?YesAdditional CommentsSupport can be asked by email to the authors as stated at the end of the README on GitHub.Is the code executable?YesAdditional CommentsThe R script can be executed using the example input files. I would suggest using the optparse library to allow users provide named arguments instead of positional arguments.Is installation/deployment sufficiently outlined in the paper and documentation, and does it proceed as outlined?YesAdditional CommentsThe example command line works well. However, the R script needs shiny and xtable packages to be loaded even if none of their functions is actually called in the script.Is the documentation provided clear and user friendly?NoAdditional CommentsA detailed documentation would improve the application proposed. In particular, more details about the different chemistries used by Illumina, MGI... and the constraints to find compatible barcodes.Is there enough clear information in the documentation to install, run and test this tool, including information on where to seek help if required?YesAdditional CommentsAvailable on the GitHub web page.Is there a clearly-stated list of dependencies, and is the core functionality of the software documented to a satisfactory level?NoAdditional CommentsThe strategy used to find barcodes seems very simple, but more details would improve the manuscript.Have any claims of performance been sufficiently tested and compared to other commonly-used packages? NoAdditional CommentsThe manuscript cites several packages developed to find compatibles sequencing barcodes but the performances are not compared. Moreover, we do not know if NucBalancer still work with a high number of samples/barcodes.Is test data available, either included with the submission or openly available via cited third party sources (e.g. accession numbers, data DOIs)?YesAdditional CommentsTest data is available on GitHub within a ZIP archive. I would suggest to make them available in a raw format (.e.g. tsv or txt).Are there (ideally real world) examples demonstrating use of the software? NoAdditional CommentsA real world example would be appreciated to illustrate the software, especially in a scenario where the other cited solutions were not able to find compatible barcodes.Is automated testing used or are there manual steps described so that the functionality of the software can be verified?NoAdditional CommentsAny Additional Overall Comments to the AuthorI would suggest the authors to improve the design of the Shiny app as (at the moment) it only runs a R script and prints the result. Moreover, I think the quality of the R code could be easily improved (e.g. loops with strange counters or comparisons with booleans).RecommendationReject (Unsound or Unusable)

---

## [Reviewer Report]

Reviewer name and names of any other individual's who aided in reviewerWen YaoDo you understand and agree to our policy of having open and named reviews, and having your review included with the published manuscript. (If no, please inform the editor that you cannot review this manuscript.)YesIs the language of sufficient quality?YesPlease add additional comments on language quality to clarify if neededIs there a clear statement of need explaining what problems the software is designed to solve and who the target audience is? NoAdditional CommentsIs the source code available, and has an appropriate Open Source Initiative license <a href="https://opensource.org/licenses" target="_blank">(https://opensource.org/licenses)</a> been assigned to the code?YesAdditional CommentsAs Open Source Software are there guidelines on how to contribute, report issues or seek support on the code?NoAdditional CommentsIs the code executable?YesAdditional CommentsIs installation/deployment sufficiently outlined in the paper and documentation, and does it proceed as outlined?NoAdditional CommentsIs the documentation provided clear and user friendly?YesAdditional CommentsIs there enough clear information in the documentation to install, run and test this tool, including information on where to seek help if required?YesAdditional CommentsIs there a clearly-stated list of dependencies, and is the core functionality of the software documented to a satisfactory level?YesAdditional CommentsHave any claims of performance been sufficiently tested and compared to other commonly-used packages? NoAdditional CommentsIs test data available, either included with the submission or openly available via cited third party sources (e.g. accession numbers, data DOIs)?YesAdditional CommentsAre there (ideally real world) examples demonstrating use of the software? YesAdditional CommentsIs automated testing used or are there manual steps described so that the functionality of the software can be verified?NoAdditional CommentsAny Additional Overall Comments to the AuthorThe authors reported a new tool for barcode sequences design. This tool is developed using R/Shiny and is available for using online. Below are my comments for further improvement of the manuscript and the tool. 1. Please provide a “load example data” button in the Shiny app. With this button, the example data can be easily loaded by the users for testing NucBalancer. 2. This URL (http://146.118.68.98:8888/) for NucBalancer should also be given in the manuscript. 3. The “Download Table” button is not working. 4. Format of the input data should be checked, as input data in wrong format caused the NucBalancer to crash. 5. The authors should compare NucBalancer with published similar tools in this field. More details are required.RecommendationMajor Revisions